
*Brief communication*:

# Storm Daniel Flood Impact in Greece 2023: Mapping Crop and Livestock Exposure from SAR

Kang He[1], Qing Yang[2], Xinyi Shen[2], Elias Dimitriou[3], Angeliki Mentzafou[3], Christina

Papadaki[3], Maria Stoumboudi[3] and Emmanouil N. Anagnostou[1]

[1]Department of Civil and Environmental Engineering, University of Connecticut, Storrs, CT 06269, USA

[2]School of Freshwater Science, University of Wisconsin-Milwaukee, Milwaukee, WI 53211, USA

[3]Hellenic Centre for Marine Research, Institute of Marine Biological Resources and Inland Waters, Anavissos,

Attica 19013, Greece

*Correspondence to*: Emmanouil N. Anagnostou (emmanouil.anagnostou@uconn.edu)

**Abstract.** For this communication, we analyzed the crop area and numbers of livestock exposed to flooding from

the historic precipitation caused by storm Daniel in central Greece on September 3–8, 2023. We derived from the

near-real-time RAdar-Produced Inundation Diary (RAPID) system an inundated area totaling 1,150 km$^2$, located

mainly in the Thessalian plain. By overlaying a land cover map on the RAPID inundation map, we found that

~820 km$^2$ (70%) of the inundated area was agricultural land. A detailed distribution map of crop type and animal

farms revealed that the crop most affected by the flooding was cotton; the inundated area of more than 282 km$^2$

comprised ~30% of the total area planted with cotton in central Greece. In terms of livestock, we estimated more

than 14,000 ornithoids and 21,500 sheep and goats were affected. Consequences for agriculture and animal husbandry in Greece are expected to be severe.

## 1. Introduction

Between September 3 and September 8, 2023, the Mediterranean region was hit by storm Daniel, an
unprecedented meteorological event. Following weeks of drought, wildfires, and intense heat [PBS, 2023; CBS, 2023a], central Greece was subjected to extreme precipitation, with the 500 mm of rainfall received in a day by some cities breaking the record of observations. According to the UK Meteorological Office, for example, the rainfall accumulation in Zagora was more than 55 times higher than the average rainfall in September (~16 mm) across Greece [CBS, 2023b]. Eventually, the torrential rain compounded major flooding in central Greece, causing
extensive regions to be inundated [FloodList, 2023; NASA, 2023]. The flooding wreaked massive destruction on infrastructure, turning streets into deadly rivers, tearing down buildings and bridges, and leaving whole villages submerged [CNN, 2023; *New York Times*, 2023]. Considered the worst rainfall event in Greece's recorded history [SkyNews, 2023], the storm was also the deadliest weather event of 2023 to date [NBC News, 2023]. At least seventeen people were killed in the country, and ten were reported dead in neighboring Bulgaria and Turkey
[CNN, 2023; CBS, 2023c]. The loss from the flooding was estimated in the billions of euros [AP News, 2023], and the European Union offered Greece €2.25 billion to recover [GreekReporter, 2023a].

Agriculture was also devastated by this historic flooding event. The Thessalian plain is Greece's main agricultural breadbasket, accounting for about 12.2% of the gross value added of the agricultural industry of Greece [Hellenic Statistical Authority of Greece, 2023]. It was the worst hit area, with livestock drowned and
entire crops of cotton, corn, tomatoes, and apples destroyed [Financial Times, 2023]. Almost 70% of the cotton crop in Thessaly was estimated to have been damaged by the floods [Hürriyet Daily News, 2023]. Furthermore, production in the region was expected to be reduced by at least 50–60%, which, in turn, is expected to reduce Greece's overall cotton production by 15–20% [GreekReporter, 2023b]. Aside from the immediate damage, the
future of cotton cultivation in Greece will be adversely affected by the large proportion of the bolls that will not

open normally [eKathimerini, 2023].

To obtain a better understanding and estimation of the widespread loss to crops and livestock from storm Daniel in Greece, a real-time and accurate assessment of exposure to the flooding is needed, especially for the Thessalian plain, where agriculture plays such an important role in the national economy. For this purpose, we sought to perform rapid assessment of flood inundation and associated flood loss and damage using the near-real-

time (NRT) flood mapping capability provided by synthetic aperture radar (SAR) satellite observations [Shen et al., 2019a].

For this brief communication, we have depicted the flood-affected areas in central Greece, particularly the agricultural and husbandry land, by combining NRT inundation extents from the near-real-time RAdar-Produced Inundation Diary (RAPID) system with Coordination of Information on the Environment (CORINE)

land cover data and a detailed map of cropping and animal holding data over the Thessalian plain.

## 2. Methodology

More than half of the Thessalian plain is covered by agricultural land, with the main crops being winter wheat, maize, alfalfa, and cotton [European Commission, 2023; Greek Payment Authority of Common Agricultural Policy Aid Schemes, 2021]. The climate is continental in the western and central parts of the plain and

Mediterranean in the east. Mean annual precipitation over the Thessaly region is about 700 mm with high spatial variability, from about 400 mm in the central plain area to more than 1,850 mm in the western mountains [FATIMA, 2020].

We extracted half-hourly precipitation data on the storm Daniel event from the Integrated Multi-satellitE Retrievals for Global Precipitation Mission (IMERG) Late Precipitation L3 V06 product with 0.1-degree spatial

resolution [Huffman et al., 2019]. The IMERG half-hourly Late Run product offers near-real-time precipitation estimates with a latency of about 14 hours after data acquisition. By combining data from passive microwave


sensors and infrared sensors, it provides half-hourly global precipitation estimates with a spatial resolution of 0.1°
x 0.1°—a balance of timeliness and accuracy that makes the product valuable for applications like flood
forecasting. We used the IMERG Late product to calculate the daily accumulated precipitation between September

3 and 8, 2023, for each grid.

We also collected precipitation observations for the same period from twenty in situ rain gauges in the
Thessalian plain, obtaining them from the WunderMap, an interactive weather map developed by Weather
Underground that provides real-time weather information (https://www.wunderground.com/wundermap). We
used the observational data to bias-adjust the IMERG precipitation data and evaluate their error. Specifically, we

used all daily accumulated precipitation data from gauges and corresponding IMERG grids to determine, first, the
overall bias of the IMERG data, using equation (1) of Table 1. We then applied the bias factor to adjust the original
IMERG data, which we hereafter call bias-adjusted IMERG precipitation. Among the error metrics we used to
evaluate the performance of the bias-adjusted IMERG precipitation were correlation coefficient (cc), bias, and
root mean squared error (RMSE), shown in equations (2), (3), and (4), respectively, in Table1.

We generated NRT inundation extents over central Greece using the RAPID system. RAPID is a fully
automated system that delineates NRT inundation extents from high-resolution (10 m) synthetic aperture radar
(SAR) imagery [Yang et al., 2021; Shen et al., 2019b]. Detailed descriptions of the RAPID algorithm and its
application to delineate the inundation area are provided by Shen et al. (2019a) and He et al. (2022).

We obtained the latest land cover map of Greece from CORINE Land Cover (CLC) inventory data

(available at https://land.copernicus.eu/pan-european/corine-land-cover/clc2018). The CLC data provide a pan-
European inventory of biophysical land cover, using a consistent classification scheme and methodology across
its member countries, and it serves as a crucial resource for environmental policy development, land use planning,
and environmental research in Europe. The five main data categories we included in this study were "artificial
surfaces," "agricultural areas," "forest and semi-natural areas," "wetlands," and "water bodies." A detailed

description     of    the    CORINE     program     and    its    nomenclature    is    provided    online    at

https://www.eea.europa.eu/publications/COR0-part1. To assess the impact of storm Daniel flooding on

agriculture and husbandry, we also used more detailed crop type and livestock distribution maps for the Thessalian

plain (Greek Payment Authority of Common Agricultural Policy Aid Schemes, 2021).

**3. Results**

Figure 1 shows the spatial distribution of the twenty gauges we used to calibrate the IMERG data and of the

accumulated precipitation from the September 3–8 heavy precipitation event. We calculated the bias adjustment

factor from equation (1) using daily accumulated precipitation from the twenty pairs of gauge-IMERG data. We

then applied the bias adjustment factor (0.41 in this study) to the original IMERG data to obtain the bias-adjusted

IMERG precipitation, which would best represent the distribution of precipitation over Greece during storm

Daniel. The precipitation we observed in central and eastern Greece (>400 mm/day) was especially heavy on

September 4, 5, and 6. The total accumulated precipitation from the event was above 600 mm (Volos, Larisa,

Trikala), as depicted by the bias-adjusted IMERG data; this total broke the record for observed precipitation in

these regions.

Figure 2 shows the validation of the bias-adjusted IMERG data against the gauge data. The line charts

of the accumulated precipitation, based on the bias-adjusted IMERG and data from four gauges (where more than

450 mm of accumulated precipitation was observed), indicate that the bias-adjusted IMERG data could

successfully capture the trends in precipitation increase, especially for the cities Volos and Karditsa and the village

Platanos, for which the precipitation amounts for the bias-adjusted IMERG and the observed IMERG were close.

The scatterplot comparing the bias-adjusted IMERG data with the gauge data shows that the former overestimated

the precipitation amount by ~84%, with the overestimation occurring mainly with low precipitation (below 100

mm/day). The cc and RMSE between the bias-adjusted IMERG data and gauge data were 0.75 and 56.55 mm,

respectively.



Figure 3 presents the inundation extents from RAPID in Greece. The RAPID inundation map is highly consistent with the precipitation map. We derived an inundated area totaling 1,150 km$^2$ from RAPID, with most

of the flooded areas found in the Thessalian plain along the Pineios river (~820 km$^2$) and associated with the more than 400 mm of accumulated precipitation that fell in these regions. We determined the inundated areas for the main land cover types by overlaying inundation extents with the CORINE land cover map. Among them, around 70% were agricultural areas (~574 km$^2$), followed by the forests (17%), wetlands (10%) and artificial surfaces (3%). The inundated agricultural areas were massively located in the Thessalian plain.

Using the detailed distribution data of crop type and livestock in the Thessalian plain, we then analyzed the flood impact on each type of crop and livestock. Figure 4 (a) displays the inundated area for each crop type as a fraction of the total land area planted with that crop in the Thessalian plain. Cotton was the most obviously affected by the flooding, with ~282 km$^2$ of its cultivation area flooded; this occupies 30.5% of the total cotton area in the Thessalian plain. The inundated area for durum wheat was 57.5 km$^2$ and for other wheats 55 km$^2$,

accounting for 11.7% and 7.1%, respectively, of the area planted with these crops. Also affected by the flooding were fodder plants and grain maize, with 30.9 km$^2$ and 23.3 km$^2$, or 4% and 12.9%, respectively, of their cultivation areas inundated. Although a relatively large portion of pasture was inundated (~19.6 km$^2$), this accounted for only 1% of the total pasture area. On the other hand, the inundated areas of seed production and industrial tomatoes were low, at 7.1 km$^2$ and 3.5 km$^2$, respectively, but they occupied 11.5% and 12% of the total

areas planted with these crops.

Figure 4 (b) shows the numbers of farms holding different types of livestock in the Thessalian plain, the numbers and percentages of those farms that were flooded by storm Daniel, and the number and percentage of each type of animal affected by flood. Among those holding ornithoids, we estimated four farms were flooded; the 14,161 animals affected comprised 1.2% of the total number of ornithoids in the Thessalian plain. One hundred

twenty-six farms holding sheep and goats were also flooded, with 21,490 (or 1.33%) of these animals affected.

Other inundated farms included fifty-eight with beehives, twenty-five with cattle, and three with pigs. Among these, 9,915 beehives (5.79%), 918 heads of cattle (0.52%), and 6,031 pigs (6.43%) were affected. The least loss occurred among rabbits and horses, with 100 and 4, respectively, affected.

**4. Closing remarks**

The unprecedented precipitation event associated with storm Daniel severely damaged agriculture and animal husbandry in central Greece. For this communication, we analyzed the flood impact in the region by overlaying the inundation extent derived from the RAPID system with CORINE land cover data and detailed distribution data on crop type and animal holding in the Thessalian plain, providing the estimated inundated area for each crop and the numbers of animals affected by the flooding. As a fully automated system, RAPID delineates NRT

inundation extents from high-resolution (10 m) SAR imagery. SAR operates in the microwave frequency range, allowing it to penetrate through clouds and acquire data both day and night. The consistent and timely data acquisition this ensures is particularly crucial during emergency flood events [Shen et al., 2019b; Hostache et al., 2018; Manavalan, 2017]. With these unique capabilities, RAPID can provide timely, accurate, and reliable flood mapping, enabling swift decision making and effective response strategies in the event of a flooding hazard [Yang

et al., 2021; Shen et al., 2019a]. The main systematic errors of the RAPID system come from the IMERG data which are used to trigger the RAPID system. IMERG data have been found to overestimate light precipitation and overestimate heavy precipitation over Greece and to produce many false alarm events [Kazamias et al., 2022; Kazamias et al., 2017].

From the RAPID inundation map, we derived an inundated area totaling 820 km$^2$ in the Thessalian plain

in central Greece. Of this, 62% (~511 km$^2$) was agricultural land, in which cotton suffered the most severe damage from flooding. The inundated area of cotton amounted to 282 km$^2$, occupying ~30% of the total area planted with that crop in central Greece. Wheat was also affected by the flooding, with 57 km$^2$ of durum wheat and 55 km$^2$ of other wheat inundated. The expected result of these impacts is a severe reduction in agricultural production in

Greece is. As for livestock, we estimated more than 14,100 ornithoids and 21,400 sheep and goats to have been

affected by the flooding, a loss that is expected to influence the country's animal husbandry economy. These

numbers account only for animals in the flooded area in the Thessalian plain; both immediate and subsequent

impacts, such as disease and lack of food, mean that the actual numbers of animals drowned or otherwise affected

by the storm throughout Greece have probably been larger.

The time needed for inundated crops to recover varies significantly, depending on such factors as crop

type, growth stage, duration of inundation, the degree of soil erosion caused by the flood, and the water quality of

the floodwaters. Cotton cultivation is generally more resilient than other crop types, especially during its growing

season. The recovery time can range from weeks to months if the inundation does not last more than a few days,

soil erosion is not significant, and plant diseases are not caused by the stagnant waters. Since cotton seeding in

Thessaly usually occurs in the spring, we can make no certain assessment at this moment of the flooding impacts

on the upcoming cultivation period there. We can say, however, that in this region, where large areas remained

inundated for several days or even weeks and the magnitude of the disaster limited opportunities to restore the

drainage network quickly, damage to specific crops—for example, tree plantations—will probably be very high,

and the consequences will be worse if most of the flooded area sees unfavorable weather conditions before the

next planting. Additional analysis is needed to assess these consequences. Field surveys will be essential to

evaluate the flood's impact on crops, considering factors such as flood depth, affected crop types (annual and

perennial), erosion, and soil composition changes. Crop damage should be classified, and the economic impact

should be estimated [AUA, 2023].

With extreme weather increasing worldwide, demands are growing for quick and accurate hazard

monitoring and prediction globally. Future directions of this study will include improving the frequency and

coverage of the NRT RAPID inundation estimates by utilizing the modern satellite constellations (for example,

ICEYE [Ignatenko et al., 2020]) and combining the estimates with developed flood models and crop data to predict



the extent of flood damage to cropland [Lazin et al., 2021] and associated socioeconomic impacts [Gould et al., 2020].

**Author contribution:** KH: Formal analysis, writing—original draft and editing. QY, ED, and AM: Software, formal analysis, data curation. CP: Writing, formal analysis. MS: review and editing. XS and EA: Conceptualization, project administration, writing—review and editing.

**Competing interests:** The authors declare they have no conflict of interest.

**Acknowledgments:** This research was supported by a National Science Foundation HDR award entitled
"Collaborative Research: Near Term Forecast of Global Plant Distribution Community Structure and Ecosystem Function." Kang He received the support of the China Scholarship Council for four years' Ph.D. study at the University of Connecticut (under grant agreement no. 201906320068).

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

Table 1. Equations of bias adjustment factor, cc, bias and RMSE

| Equation | Best value | |
|---|---|---|
| $bias\ adjustment\ factor = \dfrac{\sum(X_i * O_i)}{\sum(O_i * O_i)}$ | 1 | (1) |
| $cc = \dfrac{\sum(X_i - \bar{X})(Y_i - \bar{Y})}{\sqrt{\sum(X_i - \bar{X})^2 (Y_i - \bar{Y})^2}}$ | 1 | (2) |
| $bias = \dfrac{\sum(X_i * Y_i)}{\sum(X_i * X_i)}$ | 0 | (3) |
| $RMSE = \sqrt{\dfrac{\sum(X_i - Y_i)^2}{n}}$ | 0 | (4) |

Where $n$ is the number of data, $i$ represents an index for each individual data, $X$ is the gauge data, $O$ and $Y$ are the original and the bias adjusted IMERG data, respectively.


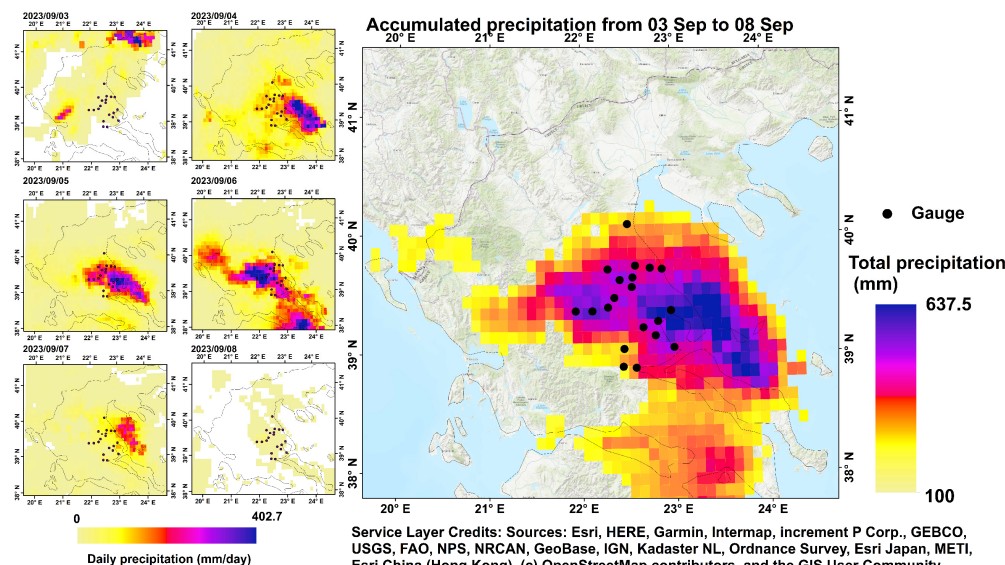

Figure 1. Spatial distribution of gauge and the daily accumulated precipitation from bias adjusted IMERG data in the period 3-8 September, 2023 during the storm Daniel in Greece.

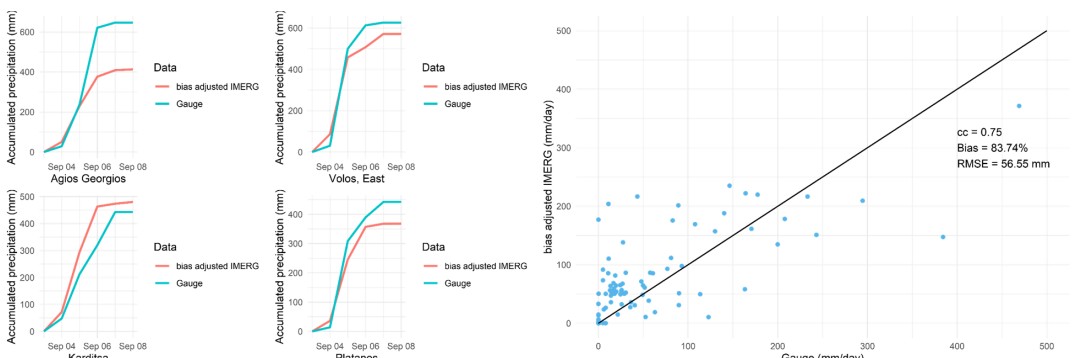





Figure 2. Left panel: Validation of the bias adjusted IMERG data against gauge data with line charts showing the accumulated precipitation from Sep 3 to 8, 2023 for 4 gauges, Agios Georgios, Volos, East, Karditsa and Platanos and Right panel: the scatterplots of the bias adjusted IMERG daily precipitation against gauges.

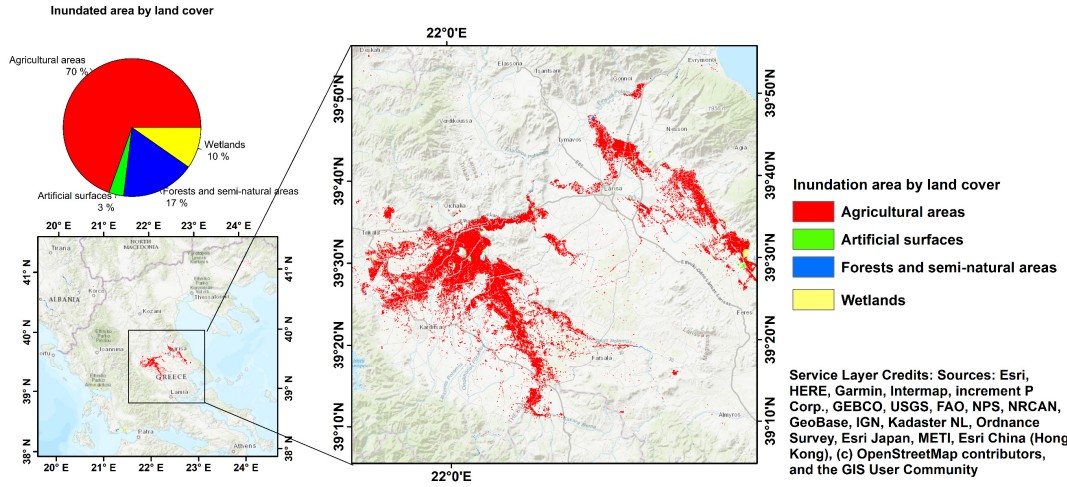

Figure 3. Inundation extends from RAPID in Greece with inundated areas for the land cover type.

*The RAPID inundation map is retrieved using the SAR images available on 06 and 12 September.*

290





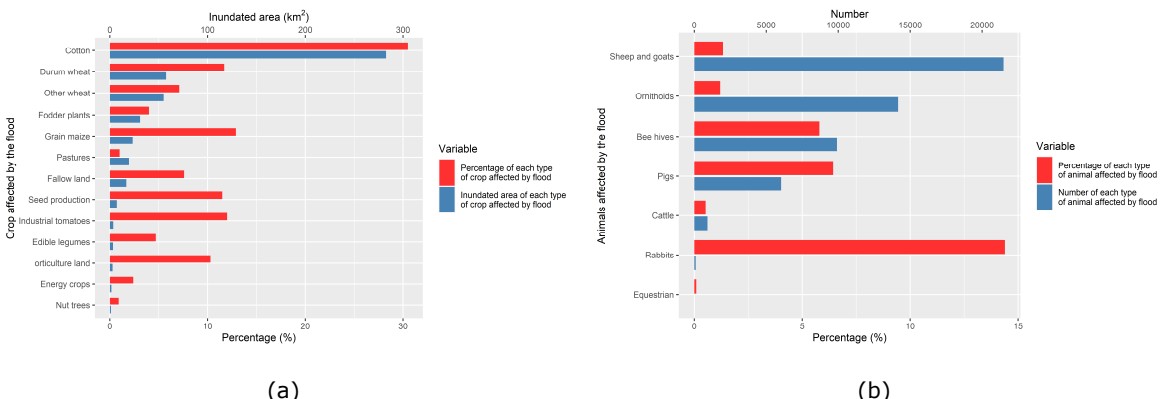

(a)          (b)

Figure 4. (a) the inundated area and percentage of each type of crop affected by flood in the Thessalian plain; (b) the number and percentage of each type of animal affected by flood in the Thessalian plain.