# Peer review of "Brief communication"

_Natural Hazards and Earth System Sciences, 2023_

## Author Response (AR1)

**Respond to reviewer #1**

This is a brief communication focused on a quick assessment of damages from the Storm Daniel Flood Impact in Greece 2023. I generally find the data and analysis are both valid. The authors may want to make more connections among precipitation assessment, inundation, and agricultural impact because they gave me an impression of three isolated items. Besides I have one major concern and a couple of minor issues for the authors to address. I want to give a minor revision to this manuscript.

**Respond**: We appreciate the reviewer's constructive comments on the manuscript to further improve the quality and the contribution of our work. Below are the authors' responses on all of the reviewer's questions and suggestions. The reviewer's comments are marked as **red**, while our responses are marked as **blue**.

Major concern: Why use the rain gauge data to correct the IMERG? There are already correction/calibration algorithms involved in IMERG. The daily recording time is different from gauge to gauge and may not be consistent. This may introduce uncertainties in gauge observations as they are used as the reference to correct IMERG.

**Respond**: First, we would like to clarify that the quality of the IMERG data does not affect the inundation results from the RAPID system, which is derived from synthetic aperture radar (SAR) images (Shen et al., 2019; Yang et al., 2021). We present the IMERG precipitation map because we want to show the spatial consistency between areas of high precipitation and areas of dense inundation. Regarding the question, we would like to note that despite the inherent correction and calibration algorithms within IMERG, IMERG is found to show bias in daily precipitation estimates over Greece (Kazamias et al, 2022), e.g. overestimation of light to moderate precipitation, and underestimation of heavy rain rates. We additionally use the rain gauge data from WunderMap (https://www.wunderground.com/wundermap) to fine tune IMERG to reflect more accurate precipitation occurring at various locations, thus showing more accurate spatial patterns of precipitation. As for the possible inconsistent recording time of gauges,, we download the hourly gauge observation  and accumulated to daily to correct IMERG. The timing difference is therefore insignificant.

References:
Yang, Q., Shen, X., Anagnostou, E.N., Mo, C., Eggleston, J.R. and Kettner, A.J., 2021. A high-resolution flood inundation archive (2016–the present) from Sentinel-1 SAR imagery over CONUS. Bulletin of the American Meteorological Society, pp.1-40. DOI:10.1175/BAMS-D-19-0319.1.

Shen, X., Anagnostou, E.N., Allen, G.H., Brakenridge, G.R. and Kettner, A.J., 2019. Near-real-time non-obstructed flood inundation mapping using synthetic aperture radar. Remote Sensing of Environment, 221, pp.302-315. DOI: 10.1016/j.rse.2018.11.008.

Kazamias, A.P., Sapountzis, M. and Lagouvardos, K., 2022. Evaluation of GPM-IMERG rainfall estimates at multiple temporal and spatial scales over Greece. Atmospheric Research, 269, p.106014. https://doi.org/10.1016/j.atmosres.2021.106014.

Minor issues

Line 110: what is "cc"?

**Respond**: "cc" represents the correlation coefficient, we have defined it in line 78 in the manuscript:

"Among the error metrics we used to evaluate the performance of the bias-adjusted IMERG precipitation were correlation coefficient (cc), bias, and root mean squared error (RMSE), shown in equations (2), (3), and (4), respectively, in Table1."

Line 134: How the 14,161 animals number was calculated is not clear, please add some details.

**Respond**: The number of animals were estimated based on the livestock farms mapped to be inundated during the flood event and the corresponding number of animals in each installation as declared to the regional offices of the Ministry of Agriculture (the responsible service is called OPEKEPE in Greek). Most of the animals in the study area would be recorded through this process since livestock breeding is a subsidized activity but there may be a number of animals not counted with this approach (eg pet animals and those living in the people's houses or in small barns). Therefore, the exact number of dead animals from the flood should be slightly higher than the number we estimated but this is the only official data that we can have.

From line 93 to line 98 in the revised manuscript:

"Overlaying the inundation map from RAPID with the distribution map of crop types, the area of crops affected by flooding was estimated, by identifying instances where the mapped crop types coincide with areas marked as inundated on the flood map. While the number of animals were estimated based on the livestock farms mapped to be inundated during the flood event and the corresponding number of animals in each installation as declared to the regional offices of the Ministry of Agriculture.

**Respond to reviewer #2**

This brief communication provides an efficient method to generate flood impact mapping using the RAPID system together with available land cover data. The methodology is valid and can be used as a tool for swift decision making.

**Respond**: We appreciate the reviewer's constructive comments on the manuscript to further improve the quality and the contribution of our work. Below are the authors' responses on all of the reviewer's questions and suggestions. The reviewer's comments are marked as **red**, while our responses are marked as **blue**.

However, the IMERG data used as input to the RAPID system seem to underestimate very heavy precipitation amounts (as those recorded during storm Daniel) even after bias correction. Can this problem be rectified , eg. by using more in situ measurement stations? Are there any more available station data (from national/regional networks) that could form a dense grid and hence generate a better bias-corrected IMERG product?

Please discuss this issue.

**Respond**: First, we would like to clarify that the quality of the IMERG data does not affect the inundation results from the RAPID system, which is derived from synthetic aperture radar (SAR) images (Shen et al., 2019; Yang et al., 2021). To show the spatial consistency between areas of high precipitation and areas of dense inundation, even biased IMERG precipitation map serves the purpose.. As a fully automated system, the role of IMERG is to trigger RAPID system to produce inundation maps when high precipitation is detected. However, the reviewer is correct, systematic errors still exist in IMERG even though some inherent correction and calibration algorithms are involved in IMERG. The bias correction of IMERG is not within the scope of this study, while some research studies have evaluated the performance of IMERG over Greece and Europe and proposed methods to improve the quality of IMERG regionally (Navarro et al., 2019; Tapiador et al., 2020; Kazamias et al., 2022; Gentilucci et al., 2022).

From line 158 to line 162 in the revised manuscript:

"Despite the inherent correction and calibration algorithms within IMERG, there is a necessity for further refinement of IMERG data to more accurately represent precipitation distribution at diverse locations [Navarro et al., 2019; Tapiador et al., 2020; Kazamias et al., 2022; Gentilucci et al., 2022)]. Such enhancements could greatly benefit from leveraging dense gauge networks, providing a more granular and precise calibration of precipitation measurements."

References:

Yang, Q., Shen, X., Anagnostou, E.N., Mo, C., Eggleston, J.R. and Kettner, A.J., 2021. A high-resolution flood inundation archive (2016–the present) from Sentinel-1 SAR imagery over CONUS. Bulletin of the American Meteorological Society, pp.1-40. DOI:10.1175/BAMS-D-19-0319.1.

Shen, X., Anagnostou, E.N., Allen, G.H., Brakenridge, G.R. and Kettner, A.J., 2019. Near-real-time non-obstructed flood inundation mapping using synthetic aperture radar. Remote Sensing of Environment, 221, pp.302-315. DOI: 10.1016/j.rse.2018.11.008.

Navarro, A., García-Ortega, E., Merino, A., Sánchez, J.L., Kummerow, C. and Tapiador, F.J., 2019. Assessment of IMERG precipitation estimates over Europe. Remote Sensing, 11(21), p.2470.

Tapiador, F.J., Navarro, A., García-Ortega, E., Merino, A., Sánchez, J.L., Marcos, C. and Kummerow, C., 2020. The contribution of rain gauges in the calibration of the IMERG product: Results from the first validation over Spain. Journal of Hydrometeorology, 21(2), pp.161-182.

Kazamias, A.P., Sapountzis, M. and Lagouvardos, K., 2022. Evaluation of GPM-IMERG rainfall estimates at multiple temporal and spatial scales over Greece. Atmospheric Research, 269, p.106014.

Gentilucci, M., Barbieri, M. and Pambianchi, G., 2022. Reliability of the IMERG product through reference rain gauges in Central Italy. Atmospheric Research, 278, p.106340.

The method presented in this brief communication provides an efficient mapping method but gives no information about crop impact and inundation duration. I recommend that additional analysis to cover this issue should be the focus of a follow-up paper.

**Respond**: Thanks for your suggestion. We have addressed this in the revised manuscript:

Form line 177 to line 180 in the revised manuscript:

"Additional analysis of flood impacts on agriculture, including crop damage and yield loss would potentially aid in more effective flood management and mitigation strategies. Besides, analyzing inundation duration through hydrological models and remote sensing could provide critical insights into flood resilience and recovery processes."